# Characterization of a *Treponema denticola* ATCC 35405 mutant strain with mutation accumulation, including a lack of phage-derived genes

Tadaharu Yokogawa[1], Keiji Nagano[2]*, Mari Fujita[2], Hiroshi Miyakawa[2], Masahiro Iijima[1]

1 Division of Orthodontics and Dentofacial Orthopedics, Department of Oral Growth and Development, School of Dentistry, Health Sciences University of Hokkaido, Hokkaido, Japan, 2 Division of Microbiology, Department of Oral Biology, School of Dentistry, Health Sciences University of Hokkaido, Hokkaido, Japan

* knagano@hoku-iryo-u.ac.jp

## Abstract

*Trepoenema denticola*, a spirochetal bacterium, is associated with periodontal diseases. The type strain of the bacterium, ATCC 35405, is commonly used in a basic research. Here, we report that our stock strain derived from ATCC 35405 had a mutation on the chromosome and expressed differential characteristics from the original strain. Genome sequencing analysis revealed the lack of a phage-derived region, and over 200 mutations in the mutant strain. The mutant grew to a higher density in broth culture as compared with the origin. In addition, the mutant formed a colony on the surface of the agar medium, whereas the origin could not. On contrary, the mutant showed decreased motility and adhesion to gingival epithelial cells. There were no differences in the bacterial cell length and a chymotrypsin-like protease activity between the two strains. RNA and genome sequencing analysis could not identify the genes that introduced the phenotypic differences between the strains. This mutant is potentially useful for examining the genetic background responsible for the physiological and pathogenic characteristics of *T. denticola*.

## Introduction

*Treponema denticola*, a gram-negative anaerobic spirochetal bacterium, is a potential pathogen associated with periodontal disease in humans [1]. This bacterium colonizes the gingival crevice through a biofilm formation with multiple bacterial species, leading to a niche of dysbiotic microbiota [2]. *T. denticola*, with its spirochetal characteristics, actively migrates in a viscous medium by rotating its body using the periplasmic flagella between the inner and outer membranes [3]. Additionally, this bacterium expresses several virulence factors such as a chymotrypsin-like protease called dentilisin, which can damage host tissues [4].

Our previous comparative analysis reported high and low motile stains in *T. denticola* [5]: highly motile strains (ATCC 33521 and ATCC 35404) diffused widely in a semisolid medium and showed vigorous rotational movement under the microscopic observation, whereas low

**Data Availability Statement:** All relevant data are within the paper.

**Funding:** This work was supported by JSPS KAKENHI (Grant Number 21K09860 (KN)).

**Competing interests:** The authors have declared that no competing interests exist.

motile strains (ATCC 33520 and ATCC 35405) did not. However, ATCC 35405, a type and genome-published strain [6], has been reported to exhibit substantial motility [7, 8]. In a previous study, we attributed the decrease in motility of ATCC 35405 to our culture method, which we developed using a novel commercially based medium [5]. However, we found that the strain showed active motility even in our culture method by using the ATCC 35405 strain freshly obtained from the RIKEN BioResource Center, a public distribution agency in Japan. Here, we report that the low motility strain derived from ATCC 35405 is a mutant lacking a possible phage-derived gene region from the original strain. Mitchell *et al.* isolated a phage particle from the ATCC 35405 strain during a log-term cultivation [9]. They also revealed that the phage genome contained a region from TDE_1133 to TDE_1173 in the bacterial genome [9]. Additionally, they detected putative *attB* and *attP* sequences, which are repeat sequences critical for integrating the phage into the bacterial genome [10], outside the bacterial genome region [9]. We demonstrated that the mutant strain isolated in this study was likely generated by prophage induction of the same phage. Additionally, we identified more than 200 short or point mutations in the mutant. We then investigated the physiological and pathogenic characteristics of the mutant.

## Materials and methods

### Bacterial strain and culture

We used two stocks originating from *T. denticola* strain ATCC 35405 and named the original (OG) and mutant (MT) strains. Although both strains were originally distributed from the RIKEN BioResource Center (Ibaraki, Japan), MT was obtained several years ago, and maintained in our laboratories and OG was obtained just before use in this study. They were cultivated in GAM broth, Modified (Nissui Pharmaceutical Co., Ltd., Tokyo, Japan) supplemented with 10 μg/mL thiamine pyrophosphate and 5% (v/v) heat-inactivated rabbit serum (mGAM-TS) at 37˚C under anaerobic conditions [11]. When needed, highly pure agar (Difco Agar Noble, Becton, Dickinson and Company, Sparks, MD, USA) was added to solidify the medium. The strains were passed through mGAM-TS solidified with 0.75% (w/v) agar in the Craigie's tube [12]. After sub-cultivation in the mGAM-TS broth, they were subjected to each experiment between the middle and late logarithmic growth phases. Bacterial growth in the broth was monitored by measuring the optical density at 600 nm (OD600).

### Genome analysis

Genomic DNA was extracted using the Wizard Genomic DNA Purification Kit (Promega Corporation, Madison, WI, USA), and was subjected to genomic DNA sequencing at a contract research organization (Veritas Genetics, Danvers, Massachusetts, USA). A sequence library was prepared using KAPA HTP Library Preparation Kits (for illumina)(Kapa Biosystems Inc., Wilmington, MA, USA). DNA sequencing was conducted using a next-generation sequencer, the NovaSeq6000 platform (150-bp paired-end reads). After the raw sequences were trimmed, and their quality was filtered (Chastity filter; Illumina, Inc., San Diego, CA, USA), and the remaining reads (7,460,000) were obtained, which provided approximately 400-fold genome coverage of the genome of ATCC 35405 (2,843,201 bp) published in GenBank (https://www.ncbi.nlm.nih.gov/genbank/). Sequence data were deposited in GenBank under accession numbers of PRJNA751694 (BioProject), SAMN20524577 (BioSample), and SRR15334816 (SRA). Gene mapping of the sequence reads on the genome of ATCC 35405 (AE017226.1 of accession number in GenBank) was performed using programs provided by BaseSpace Apps on the Illumina website (Illumina, Inc.).

## Motility assay

Bacterial motility was examined using two distinct methods. In the first method, a soft agar medium was used [13]. Briefly, *T. denticola* cultures (1 μL) were carefully placed on semi-solid mGAM-TS containing 0.5% agar. After anaerobic incubation at 37˚C for 5 days, turbid plaques were measured. In the second method, the bacterial culture was observed under a phase-contrast microscope at 37˚C [5]. Videos of this motion, recorded on a computer, were played in slow motion using a Windows Media Player (Microsoft, Redmond, WA, USA), and rotational rates were measured.

## Transmission electron microscopy

Bacterial cultures were washed once in phosphate-buffered saline (PBS), pH 7.4, placed on a carbon support film grid, negatively stained with ammonium molybdate (containing 5 mM molybdenum), pH 7.0, and observed by using the JEM1400 Plus Electron Microscope (JEOL, Tokyo, Japan). Bacterial cell lengths were measured by tracing the microphotographs.

## Chymotrypsin-like protease activity assay

The assay was performed as previously described [14]. Briefly, the synthetic chromogenic substrates for chymotrypsin [N-succinyl-L-alanyl-L-alanyl-L-prolyl-L-phenylalanine 4-nitroanilide (SAAPFNA)] was obtained from Sigma-Aldrich Co. LLC (St. Louis, MO, USA). The substrate releases 4-nitroaniline by enzymatic cleavage, and the concentration of 4-nitroaniline is measured at OD410. Bacterial suspensions, in 50 mM Tris-HCl, pH 7.5, supplemented with 2 mM dithiothreitol and 150 mM NaCl, were adjusted to 0.2 of OD600 [corresponding to $10^9$ cells/ mL of *T. denticola* [11]]. The substrate was prepared at 2 mM in the same buffer used for the bacterial suspensions. The bacterial suspension and the substrate were preincubated at 37˚C, mixed in equal volumes (150 μL each), and incubated at 37˚C for 30 min. The reaction was stopped with the addition of 75 μL of 25% (v/v) acetic acid. After the bacterial cells were removed by centrifugation, measurements at OD410 was performed using a 1-cm path length cuvette. Protease activity was calculated using the molar extinction coefficient of the substrate as 8,800 $M^{-1}$ $cm^{-1}$ at 410 nm, pH 7.5.

## Infection of gingival epithelial cells with *T. denticola*

We performed an *in vitro* infection assay under anaerobic conditions as described previously [5]. Briefly, the human gingival epithelial cell line Ca9-22 (RIKEN BioResource Center) was maintained in Dulbecco's Modified Eagle's Medium (DMEM, Cat #21063–029, Life Technologies Corporation, Carlsbad, CA, USA) supplemented with 10% heat-inactivated fetal bovine serum at 37˚C under 5% $CO_2$. Confluent Ca9-22 cells ($2 \times 10^6$ cells per insert, 0.47-cm$^2$ culture area) were infected with $2 \times 10^8$ *T. denticola* cells, corresponding to a multiplicity of infection (MOI) of 100, in 0.4 mL, and then incubated under anaerobic conditions. Notably, the bacterial count in the medium changed little during the 3 h of the experiment. In addition, OG maintained active motility throughout the experimental period. Thereafter, the insert was washed thoroughly with PBS, pH 7.0, to remove planktonic bacteria and soaked in 4% (w/v) paraformaldehyde in PBS, pH 7.0 to fix the cells. The fixed sample was washed with PBS and then permeabilized with PBS containing 0.1% Triton X-100 at 37˚C for 30 min. The sample was washed again and then blocked with PBS containing 3% bovine serum albumin at room temperature for 60 min. The cells were then incubated with anti-whole *T. denticola* cell rabbit serum (1:1,000 dilution) for 30 min at room temperature. After washing with PBS, the samples were simultaneously incubated with Alexa Fluor 488-conjugated goat IgG fraction to the

rabbit IgG secondary antibody (1:1,000 dilution; Life Technologies Corporation) and Alexa Fluor 568-conjugated phalloidin (1 µg/mL; Life Technologies Corporation) for 60 min at room temperature in the dark to label the bacterial cells and actin filaments of Ca9-22 cells, respectively. After thorough washing with PBS, the bottom membrane of the insert was excised, mounted with ProLong Gold antifade reagent (Thermo Fisher Scientific Inc., Rockford, IL, USA), and observed by confocal laser scanning microscopy (CLSM, Zeiss LSM 710, Carl Zeiss AG Co., Oberkochen, Germany). *T. denticola* cells were counted in the captured images with a visual field of 130 µm × 130 µm (= 16,900 µm$^2$). We expressed the bacterial counts as the association number including adherence to the cell surface and invasion into the cell.

## Quantitative analysis of transcription

A quantitative analysis of transcription was performed using RNA sequencing (RNA-Seq). Briefly, bacterial cultures, at 0.03 to 0.04 OD600 (in the middle logarithmic growth phase), were immediately collected by centrifugation at 4˚C. Total RNA was isolated using ISOGEN (Nippon Gene Co., Ltd., Tokyo, Japan), and treated with RNase-free recombinant DNase I (Takara Bio, Inc., Shiga, Japan) to remove residual DNA strands. RNA-seq was performed using the Illumina NovaSeq6000 system (Illumina, Inc., San Diego, CA, USA) at a contract research organization (Filgen Inc., Nagoya, Japan). The raw sequence data were analyzed by a comprehensive analysis of software for next-generation sequencing using CLC Genomics Workbench ver. 11 (Qiagen, Hamburg, Germany). The procedure was performed twice using a different bacterial culture lot. Sequence data were deposited in GenBank under accession numbers of PRJNA831582 (BioProject) and PRJNA831582 (SRA). BioSample accessions are SAMN27755115 and SAMN27755116 for OG and MT, respectively.

## Protein electrophoresis and mass spectrometry analysis

The bacterial cells were lysed in BugBuster HT (EMD Millipore Co., San Diego, CA, USA), and protein concentrations were measured using the bicinchoninic acid method (Pierce BCA Protein Assay Kit, Thermo Fisher Scientific Inc.). The lysates were mixed with a loading buffer consisting of 50 mM Tris-HCl, pH 6.8, 1% (w/v) sodium dodecyl sulfate (SDS), 0.5 M 2-mer-captoethanol, 10% (w/v) glycerol, and 0.01% bromophenol blue (all at final concentrations), and denatured by heating at 100˚C for 5 min. Then, the samples were loaded onto an SDS-polyacrylamide gel electrophoresis (PAGE) gel consisting of 11% polyacrylamide. After electrophoresis, protein bands were visualized by staining with Coomassie Brilliant Blue R-250 (CBB). Proteins detected by SDS-PAGE and CBB staining were identified by matrix-assisted laser desorption/ionization-time-of-flight mass spectrometry (MALDI-TOF MS) [15]. Briefly, after in-gel tryptic digestion, the peptides were analyzed using a 4800 MALDI TOF/TOF Analyzer or 4800 Plus MALDI TOF/TOF Analyzer (AB Sciex, Framingham, MA, USA). Protein identity was determined based on raw MS/MS data from one or more peptides using an MS/MS Ion Search of Mascot database search (Matrix Science Inc., Boston, MA, USA).

## PCR and DNA sequencing

Primer sets for PCR were designed based on the sequence information of the ATCC 35405 strain published in GenBank: 5ʹ-CCATGCTCAACAATATCTCCCATTG-3ʹ/5ʹ-CACTTAG TTTTCCTCCTGCCTCC-3ʹ, and 5ʹ-GAGGAGGAGGATAAGCTATCCG-3ʹ/5ʹ-GGGATAG ACCTTTCTCAGCC-3ʹ for amplification of an inter region of the TDE_1698 and TDE_1699 genes, and a region within the TDE_2603 gene, respectively. The PCR products were subjected

to dye terminator sequencing at a contract research organization (Eurofins Genomics K.K., Tokyo, Japan).

### Statistical analysis

Data are presented as the mean ± SD. The Student's *t*-test and one-way analysis of variance followed by the Student-Newman-Keuls (SNK) test were used to analyze of differences between two and more than two groups, respectively. A statistical analysis of the transcriptome was performed based on reads per kilobase of transcript per million mapped reads (RPKM) in the CLC Genomics Workbench. A value of $P < 0.01$ was considered statistically significant.

## Results

### Genome analysis

Genome analysis using next-generation sequencing revealed more than 200 mutations in the MT strain (S1 File). Notably, an approximately 40-kbp region, including the TDE_1133 to TDE_1173 genes, was deleted. This region was consistent with the phage genome released from the ATCC 35405 strain [9]. The putative *attP*/*attB* repeat sequence (AAGCCCTATTGC CGCTA) [9] was detected only at the upstream of TDE_1173, but not at the downstream of TDE_1133 in the MT genome. However, we discovered another repeat sequence (TTTAAA GGCACTCAAAWGAGTGCCTTTTTTTA) on both sides in the MT genome (S2 File). Additionally, the analysis detected two long deletions, and dye-terminator DNA sequencing precisely determined them: 558-bp and 266-bp regions between TDE_1698 and TDE_1699, and within TDE_2603, respectively (S1 File). Furthermore, we detected five short (2–13 bp) and 233 point mutations (S1 File). Among them, 203 mutations existed within the open reading frame, of which 72 frameshift and nonsense mutations. A missense mutation (alanine to threonine at the 186th amino acid residue) in the TDE_2766 gene was annotated as coding for the flagellar motor MotA. Additionally, two genes (TDE_0169 and TDE_0647) were annotated as coding for a chemotaxis protein with missense and frameshift mutations, respectively. Furthermore, TDE_0405 and TDE_1072, which encode a major surface protein, contained one and three missense mutations, respectively.

### Comparative analysis of physiological and pathogenic characteristics

Both strains showed a similar growth rate at the early stage, until 12 h, in the mGAM-TS broth (Fig 1A). However, OG reached a plateau at an OD600 of 0.2, while MT continued to grow until an OD600 of approximately 1.0. Additionally, MT formed a colony on the surface of the medium solidified with 3% agar, but OG grew insignificantly (Fig 1B). OG showed a larger turbid ring than MT in the soft agar medium containing 0.5% agar (Fig 2A and 2B). Phase-contrast microscopy revealed that 59.4% of OG showed active rotational motility (>5 Hz), and the rest showed moderate motility (1–5 Hz). In contrast, MT showed that only 9.1% cells exhibited active motility, with most showing moderate motility (Table 1). No statistically significant difference was observed in the cell length between the two strains (Table 2). Two flagellar filaments were observed from one end in most of the cells in both OG and MT (S1 Fig). The numbers indicate the percentage (%) of bacterial cells showing a rotational motility of less than 1, 1 to 5, and more than 5 Hz. This experiment was executed three times in different cultures, and, totally 293 and 438 bacterial cells of OG and MT, respectively, were examined.

No statistically significant difference was observed in the chymotrypsin-like protease activity between the two strains (Table 3). The bacterial numbers associated with epithelial cells did not differ between the strains until 1 h post-inoculation (Fig 3). However, the association

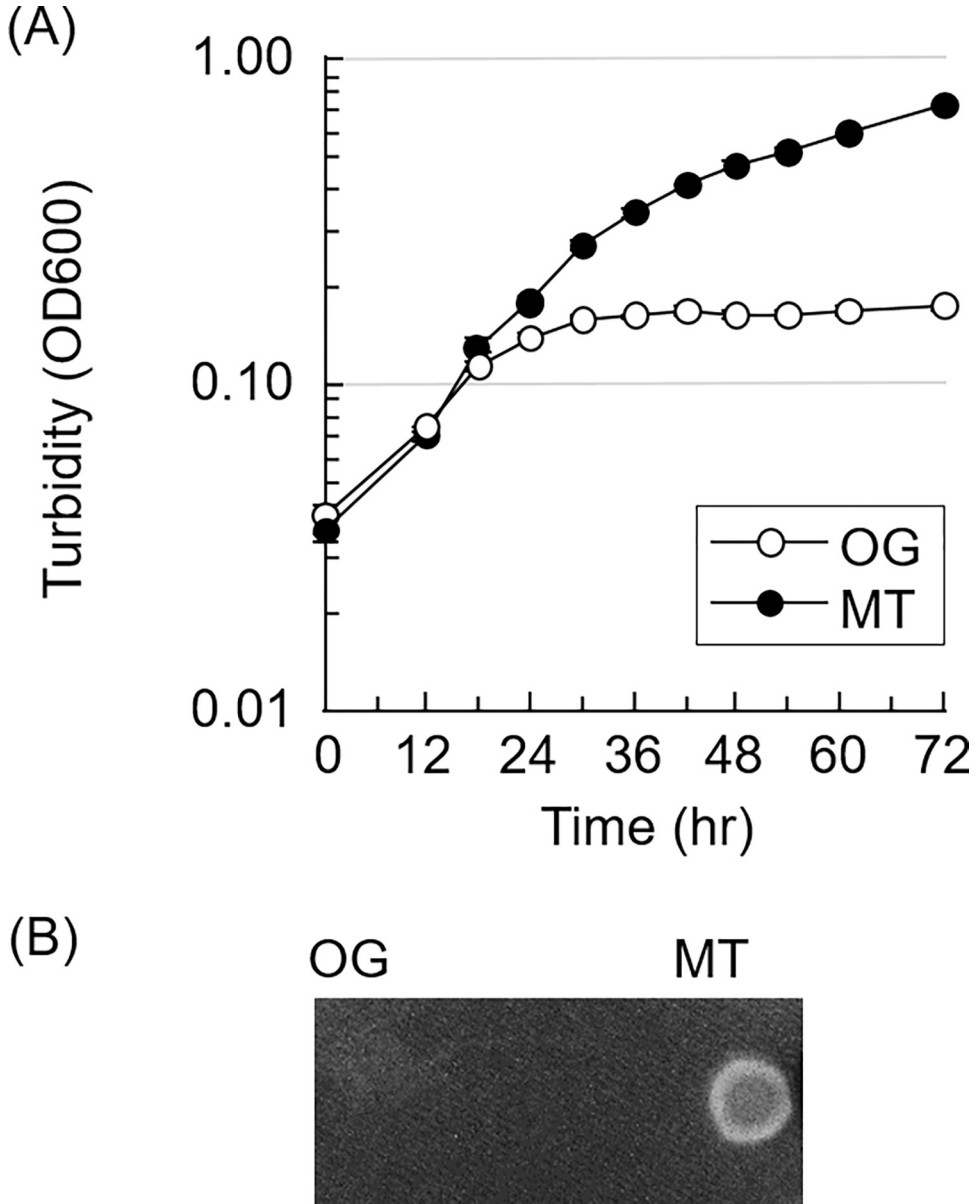

**Fig 1. Growth in the liquid and solid media.** (A) Turbidities (OD600) of the original (OG) and mutant (MT) strains in the broth were monitored at 37˚C under the anaerobic conditions. Each symbol shows the mean in quadruplicate. Note that the standard deviations are almost hidden by symbols. (B) A typical colonies of the OG and MT strains on the medium solidified with 3% agar. They were inoculated on the surface of the solid medium, and incubated at 37˚C under the anaerobic conditions for a week.

numbers of OG increased markedly at 2 and 3 h post-inoculation and was statistically significantly higher than that of MT.

## Transcriptional analyses

Transcriptional analysis was performed using two independent RNA sequencing experiments (S3 File). Table 4 lists the genes with a statistical significance ($P < 0.01$) and fold change greater than the absolute value of 2. MT showed significantly higher and lower transcriptional activity

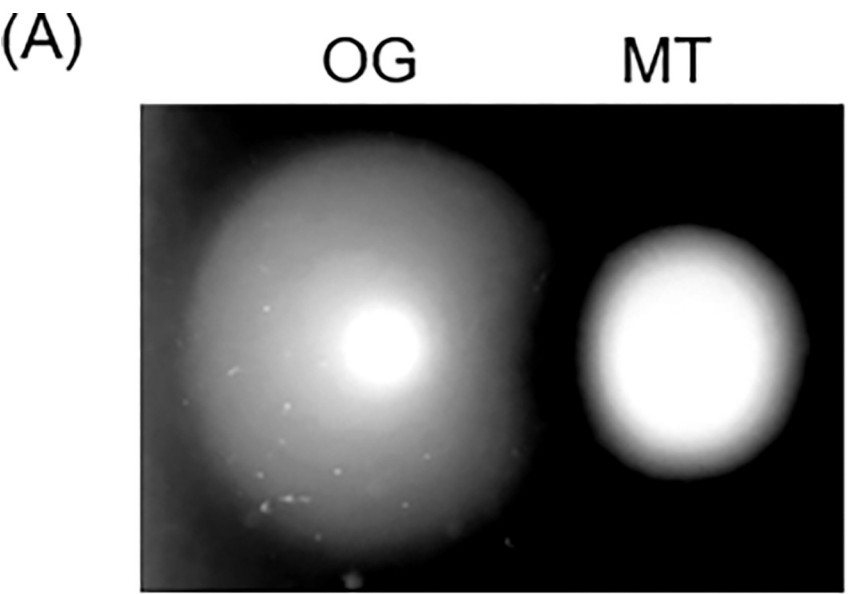

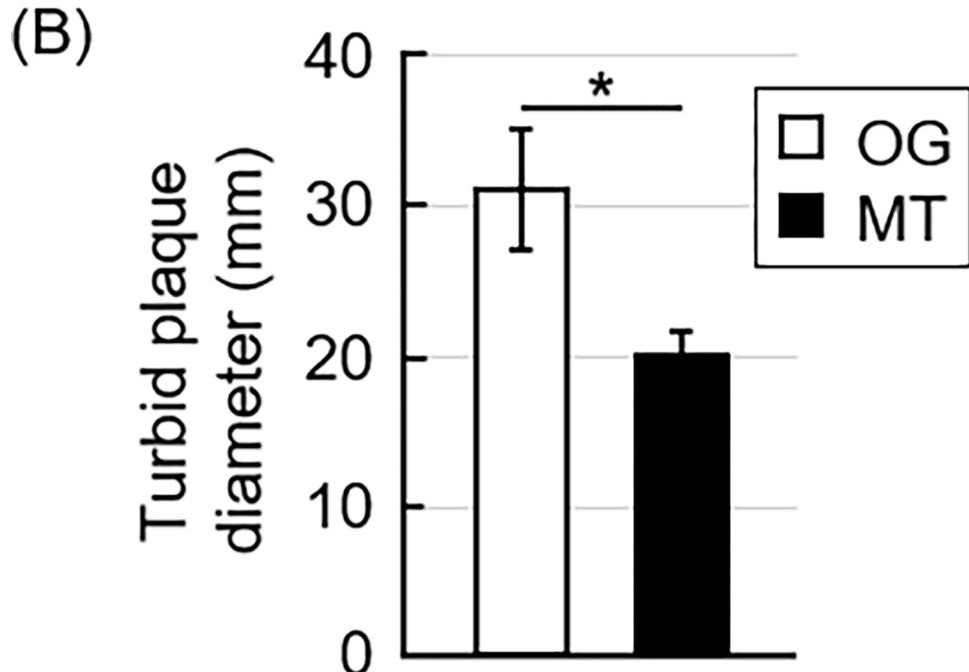

**Fig 2. Motility assay.** The original (OG) and mutant (MT) strains were inoculated on the surface of the medium containing 0.5% agar and incubated for five days at 37°C under the anaerobic conditions. A typical result is shown in panel A. This experiment was performed three times in different cultures, each in quadruplicate, and the mean and SD are shown in panel B. *Statistically significant difference between the strains ($P < 0.01$).

in 10 and 37 genes, respectively, compared to that in OG (Table 4). Among the 10 genes with higher transcriptional activity in MT, seven genes were annotated with a transporter. In contrast, 10 genes with similar annotations were also detected in the genes with lower transcriptional activity in MT. The TDE_1072 gene showed the highest fold change value, and an

**Table 1. Motility of the original (OG) and mutant (MT) strains under the phase-contrast microscopic observation.**

| Rotation rate (Hz) | OG | MT |
|---|---|---|
| < 1 | 0 | 2.3 |
| 1–5 | 40.6 | 88.6 |
| > 5 | 59.4 | 9.1 |

increase in protein expression was also confirmed (Fig 4). Two genes (TDE_2142 and TDE_0181), which were annotated with methyl-accepting chemotaxis proteins, were lower in MT than in OG.

## Discussion

We revealed that the MT strain derived from *T. denticola* ATCC 35405 lacked the 40-kbp long region, including TDE_1133 to TDE_1173 in the OG strain. The lack of a region in the MT genome was consistent with that reported by Mitchell *et al.* [9]. Although the putative *att* sequence [9] remained only on one side of the MT genome, another repeat sequence was detected on both sides of the lacking region (S2 File). These results demonstrate that the MT strain was generated by the spontaneous loss of the phage region during the passage process. No other regions were found in the phage-related genes accumulated in the genome of the ATCC 35405 original strain (published in GenBank). However, two genes, TDE_1209 and TDE_2742, were annotated as the phage integrase family site-specific recombinase. Additionally, two long DNA regions, from TDE_1698 to TDE_1699 (558 bp) and within TDE_2603 (266 bp), were missing from the MT chromosome. However, we did not notice any signs of a phage or transposon, which caused gene reduction. Moreover, over 200 short and point mutations were identified. We discovered 72 frameshift and nonsense mutations that might cause a deficiency in protein expression. However, we found seven frameshift mutations in the MT genome that were not presumed to express a protein in the ATCC 35405 genome due to a possible frameshift mutation. These proteins are likely expressed in MT. This suggests that a few sequencing errors remain in the ATCC 35405 genome data.

MT grew up to a much higher density in the broth media than OG did. Additionally, the high growth activity of MT was demonstrated by a colony formation on the 3% agar-containing medium, although *T. denticola* does not grow on the agar media due to inhibition of growth by agar or components contained in the agar [16, 17]. Notably, *T. denticola* does not form a colony on the surface of the medium solidified with 1.5% agar, and diffuses inside the medium. Although the genome analysis did not reveal a genetic background to facilitate the growth of MT, RNA sequencing revealed differential expression of genes involved in a transporter between the strains. These transporters may play a role in nutrient uptake [18]. Although both increased and decreased transcription of transporter-related genes were observed, TDE_1072, which was presumed to be a major transporter in this bacterium [Fig 4 and [11]], was significantly upregulated in MT, prompting us to investigate the function of the TDE_1072 protein in nutrient uptake. Moreover, the genes TDE_1073 to TDE_1076, also

**Table 2. Bacterial cell length (μm) of the original (OG) and mutant (MT) strains.**

| OG | MT |
|---|---|
| 7.1 ± 1.79 | 7.0 ± 1.49 |

Mean ± SD (n = 30).

**Table 3. Chymotrypsin-like protease activity (/μM/cm/cell) of the original (OG) and mutant (MT) strains.**

| OG | MT |
| --- | --- |
| 6.759 ± 0.213 | 6.913 ± 0.343 |

Mean ± SD (n = 3).

annotated with a transporter, were also upregulated in the transcription, suggesting that these genes (TDE_1072 to TDE_1076) form an operon, and proteins encoded by these genes might function cooperatively as active transporters. Additionally, we should note that three missense mutations accumulated in the TDE_1072 gene of MT, which might confer a functional alteration.

MT exhibited perceptibly smaller turbid plaques in 0.5% agar-containing medium than OG, indicating a decrease in motility of MT. We also examined the motility under microscopic observation because diffusion in agar medium also depends on growth activity. MT exhibited a significantly lower motility than OG did. Genome sequencing detected a missense mutation (from alanine to threonine at the 186th amino acid residue) in TDE_2766, which encodes fla-gellar motor MotA. The effect of this mutation on the bacterial motility should be examined using site-specific mutagenesis. We also found mutations in two genes, TDE_0169 and TDE_0647, which are annotated as coding for a chemotaxis protein [19]. Additionally, RNA sequencing detected a significant decrease in the transcription of TDE_0181 and TDE_2142, which were annotated as methyl-accepting chemotaxis proteins. The lower motility of MT may be attributed to these mutations and transcriptional repressions. However, these genes do not seem to encode a major chemotaxis protein because 23 genes were annotated with the che-motaxis protein in the ATCC 35405 genome. Eight genes among them (TDE_0347, TDE_2549, TDE_1009, TDE_1492, TDE_1589, TDE_1491, TDE_2270 and TDE_1493) showed RPKM (an index of transcriptional activity) values 1.7 to 6.7 times higher than the four genes of TDE_0169, TDE_0181, TDE_0647, and TDE_2142 in OG (S3 File), suggesting that the proteins encoded by these four genes do not function as major chemotaxis proteins. Unfortunately, we did not clarify the genetic background leading to a decrease in motility of MT. It is necessary to further analyze absent genes and those with altered transcriptional activity. Additionally, we should note that bacterial motility is associated with growth activity [20, 21].

The bacterial numbers of the two strains associated with epithelial cells were comparable up to 1 h post-inoculation, but OG showed a significantly higher number than MT after 2 h. OG maintained motility over the 3-hr experiment. Our previous report showed that low motility strains tended to show higher adherence to the epithelial cells [5]; however, this study showed the opposite result. Factors other than motility were possibly involved in the bacterial adher-ence in the previous report because we used different strains with different genetic back-grounds. Although we did not define any genes involved in adherence, lower motility does not always result in the higher adherence activity. Major outer membrane protein (Msp) of *T. den-ticola* largely functions in adhesion to epithelial cells [22–24]. However, RNA sequencing did not detect a significant difference in the transcription of the *msp* gene between the two strains. Additionally, genome sequencing detected the replacement of glutamic acid with lysine at the 519th amino acid residue in Msp. We must further examine the involvement of the mutation in the functional alteration of Msp.

This study also showed that there might be several steps in adherence to epithelial cells in *T. denticola* because the difference in the bacterial numbers associated with the epithelial cells became significant after 2 h post-inoculation (Fig 3). Pathogenic *Escherichia coli* adheres to

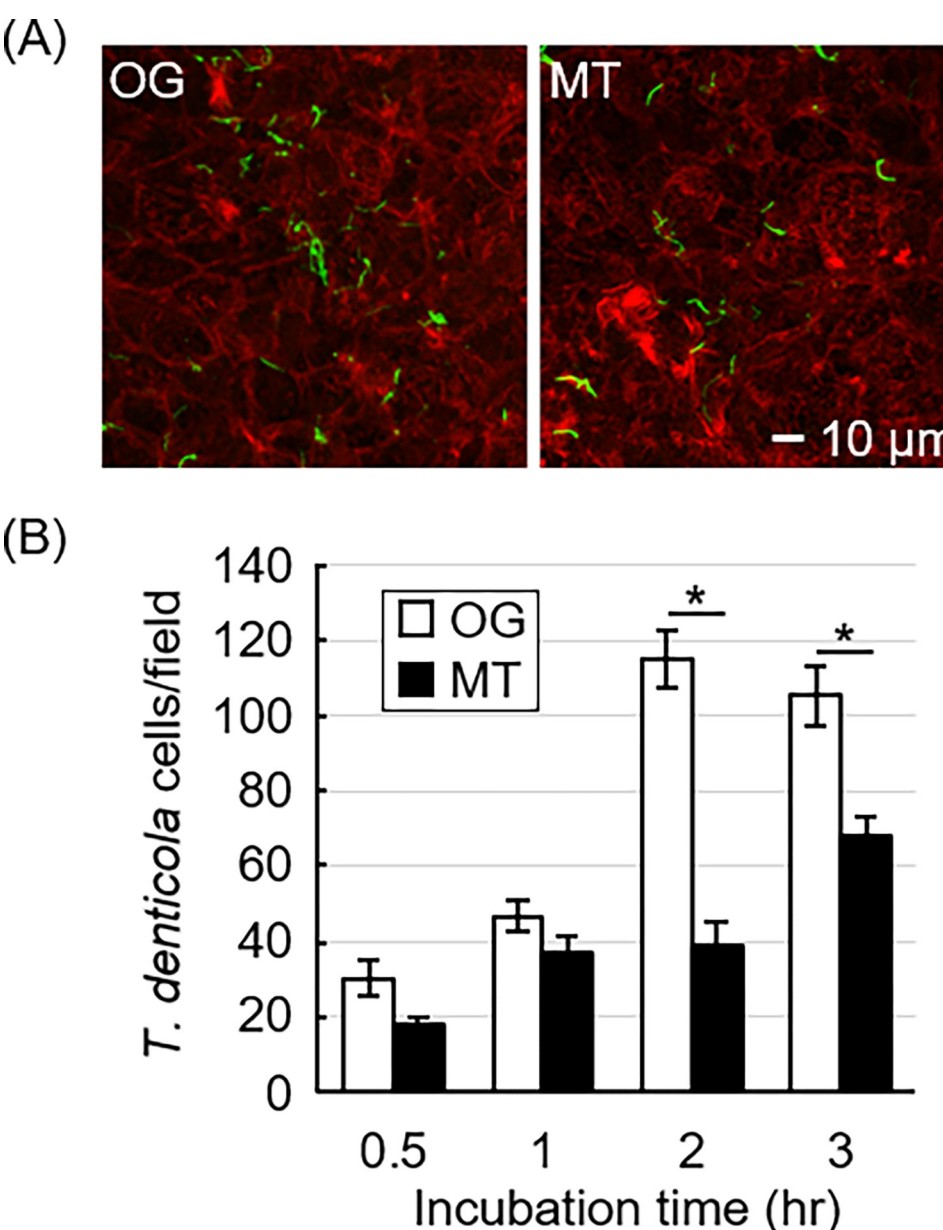

**Fig 3. Association of the original (OG) and mutant (MT) strains with gingival epithelial cells.** The strains (multiplicity of infection of 100) were incubated with confluent Ca9-22 cells for 0.5, 1, 2, and 3 h under anaerobic conditions, and then visualized by fluorescent staining. A representative image of *T. denticola* (green) associating with Ca9-22 cells (red) at 3 h post-inoculation is shown in panel A. *T. denticola* cells associating with the epithelial cells were counted in a visual field of 16,900 $\mu m^2$, and the numbers were expressed as the means ± SD in panel B. This experiment, with 30 images counted for each strain, was performed three times independently. *Statistically significant difference between the strains ($P < 0.01$).

epithelial cells through two distinct mechanisms: initial attachment by pili and subsequent adhesion by type III secretion system [25]. The difference in adherence behavior between OG and MT suggests the presence of multiple adherent molecules that function at different stages.

Quorum sensing and two-component regulatory systems are involved in the control of bacterial growth and motility [26]. MT may also be a useful tool in such analyses.

**Table 4. Genes with significantly different transcriptional activity between the original and mutant strains.**

| Locus tag | Fold change | Annotation |
|---|---|---|
| 1072 | 5.68 | ABC-type nickel/oligopeptide-like protein |
| 1075 | 5.26 | Oligopeptide/dipeptide ABC transporter |
| 1073 | 4.86 | Oligopeptide/dipeptide ABC transporter |
| 1074 | 4.45 | Oligopeptide/dipeptide ABC transporter |
| 2033 | 4.22 | ISTde1, transposase |
| 1076 | 4.01 | Oligopeptide/dipeptide ABC transporter |
| 2336 | 3.64 | Sodium/dicarboxylate symporter family protein |
| 487 | 3.45 | ABC transporter ATP-binding protein |
| 486 | 2.91 | Membrane protein, putative. ABC-2 type transporter |
| 1688 | 2.56 | Membrane protein, putative |
| 957 | -2.03 | Glycerophosphoryl diester phosphodiesterase family protein |
| 1478 | -2.06 | Conserved hypothetical protein |
| 2125 | -2.31 | Aat (leucyl/phenylalanyl-tRNA-protein transferase) |
| 988 | -2.74 | Oligopeptide/dipeptide ABC transporter |
| 1660 | -2.75 | Leucine rich repeat protein |
| 2565 | -2.98 | Hypothetical protein |
| 1946 | -2.98 | Conserved hypothetical protein |
| 1506 | -2.98 | SdhA (L-serine dehydratase) |
| 1945 | -3.05 | DsrE/DsrF-like family protein |
| 1948 | -3.09 | ABC transporter permease |
| 610 | -3.16 | 3-hydroxyacyl-CoA dehydrogenase |
| 867 | -3.25 | Hypothetical protein |
| 2239 | -3.50 | Formylglycine-generating enzyme |
| 987 | -3.55 | Oligopeptide/dipeptide ABC transporter |
| 1947 | -3.72 | ABC transporter, permease protein |
| 180 | -3.78 | Hypothetical protein |
| 2141 | -4.03 | Hypothetical protein |
| 121 | -4.06 | Lipoteichoic acid synthase |
| 704 | -4.34 | SPFH domain |
| 2546 | -4.34 | Uncharacterized membrane-anchored protein |
| 181 | -4.52 | Methyl-accepting chemotaxis protein |
| 1507 | -4.53 | SdhB (L-serine dehydratase) |
| 29 | -4.73 | ABC transporter, ATP-binding protein |
| 28 | -5.31 | ABC transporter, ATP-binding protein |
| 986 | -5.51 | Oligopeptide/dipeptide ABC transporter ATP-binding protein |
| 2142 | -5.71 | Methyl-accepting chemotaxis protein |
| 2639 | -5.78 | PepF (oligoendopeptidase) |
| 2231 | -6.42 | Internalin-like protein |
| 983 | -6.43 | Oligopeptide/dipeptide ABC transporter |
| 984 | -6.58 | Oligopeptide/dipeptide ABC transporter |
| 985 | -7.06 | Oligopeptide/dipeptide ABC transporter |
| 120 | -7.60 | Gamma-glutamyl ligase |
| 2088 | -8.72 | RNA polymerase-binding transcription factor |
| 2267 | -8.73 | Helicase and RNase D C-terminal domain protein |
| 2279 | -9.22 | Histidine kinase-like ATPases |
| 2547 | -36.75 | Hypothetical protein |
| 2269 | -52.23 | Formylglycine-generating enzyme |

The first letters (TDE_) are omitted from the locus number.

Negative and positive values indicate that the original and mutant strains, respectively, were predominantly detected.

Genes with a fold change greater than the absolute value of 2, and with statistical significance ($P < 0.01$) are shown.

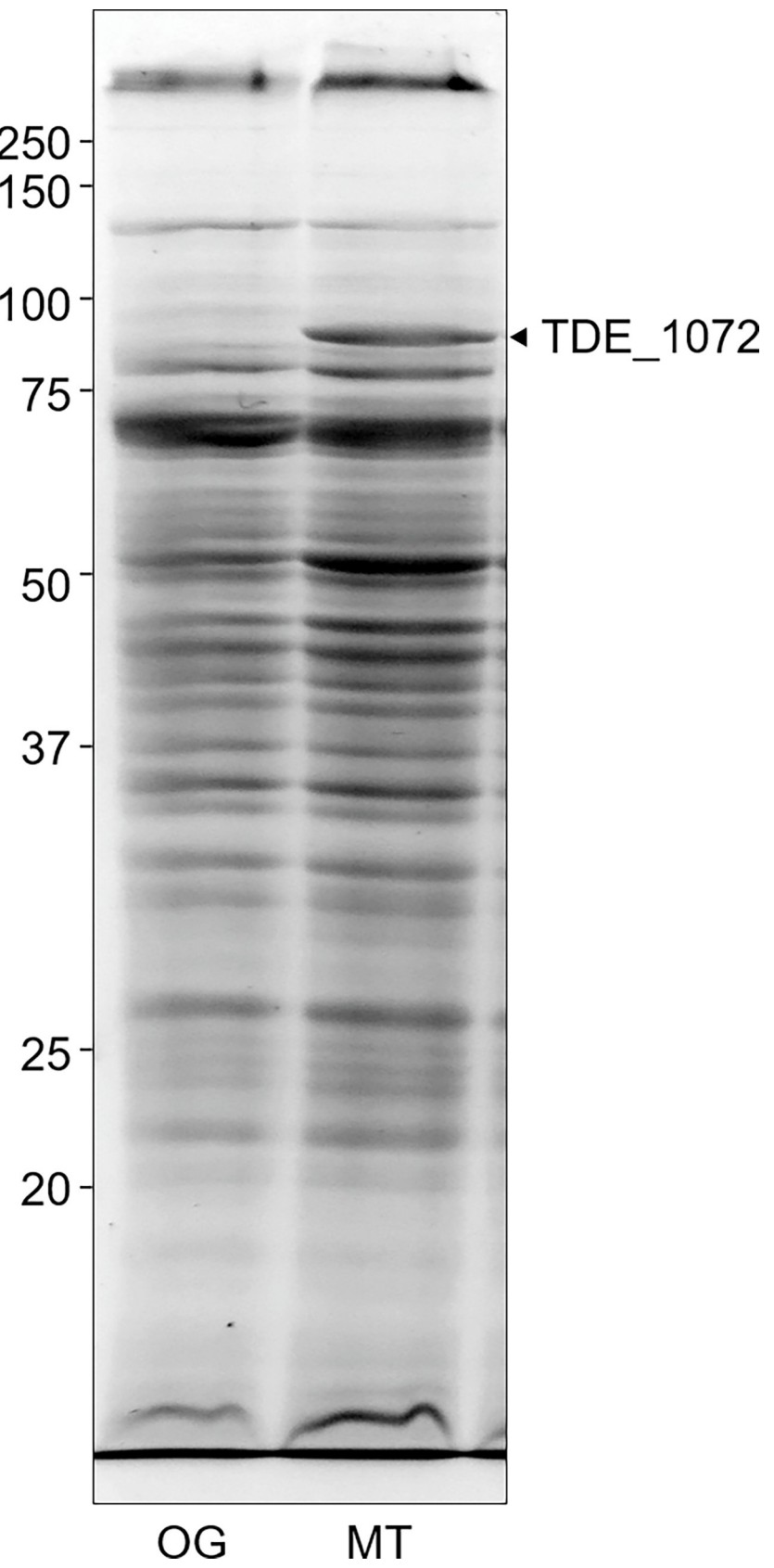

**Fig 4. SDS-PAGE and mass spectrometry analyses.** Bacterial lysates (50 μg) of the original (OG) and mutant (MT) strains were subjected to an SDS-PAGE analysis, followed by CBB staining. A major band (arrowhead), which is clearly detected in MT, was identified as the TDE_1072 protein by mass spectrometry. Molecular-weight standards (kDa) are shown on the left.

There were no differences between the strains in terms of cell length and chymotrypsin-like protease activity, suggesting that these features in *T. denticola* were not affected by the genes missing in MT.

## Conclusion

The MT strain accumulating more than 200 mutations including a lack of a phage-derived region showed a significant alteration in growth, motility, and adhesion to epithelial cells. This strain may be potentially useful for examining the genetic background responsible for the physiological and pathogenic characteristics of *T. denticola*.

## Supporting information

**S1 Raw image.**
(PDF)

**S1 File. List of mutations in the MT strain.** Mutations were detected by mapping the short-read sequences of the MT strain onto the genome of the ATCC 35405 (OG) strain published in GenBank (accession number AE017226.1).
(XLSX)

**S2 File. DNA sequences around TDE_1173 and TDE_1133.** DNA sequences around TDE_1173 and TDE_1133 of the original strain (ATCC 35405), a region detected in the MT strain (remaining sequence), the putative *att* sequence, and another repeat sequence.
(DOCX)

**S3 File. RNA sequencing data.** Results of two independent RNA sequencing experiments of OG and MT and their statistical analyses.
(XLSX)

**S1 Fig. Transmission electron microscopy.** Two flagellar filaments were transparently observed from the end of the cells in both OG and MT.
(TIF)

## Acknowledgments

We thank Mariko Kondo (Aichi Gakuin University) for helping with the experiments, Filgen Inc., for their instruction on the RNA sequence analysis, and Editage (www.editage.com) for English language editing.

## Author Contributions

**Conceptualization:** Keiji Nagano.

**Data curation:** Keiji Nagano, Mari Fujita, Hiroshi Miyakawa, Masahiro Iijima.

**Formal analysis:** Keiji Nagano, Mari Fujita, Hiroshi Miyakawa.

**Funding acquisition:** Keiji Nagano.

**Investigation:** Tadaharu Yokogawa, Keiji Nagano, Mari Fujita.

**Methodology:** Tadaharu Yokogawa, Keiji Nagano.

**Project administration:** Keiji Nagano.

**Resources:** Keiji Nagano.

**Software:** Keiji Nagano.

**Supervision:** Keiji Nagano.

**Validation:** Keiji Nagano, Mari Fujita, Masahiro Iijima.

**Visualization:** Keiji Nagano.

**Writing – original draft:** Tadaharu Yokogawa, Keiji Nagano, Mari Fujita, Hiroshi Miyakawa, Masahiro Iijima.

**Writing – review & editing:** Tadaharu Yokogawa, Keiji Nagano, Mari Fujita, Hiroshi Miyakawa, Masahiro Iijima.

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
