## [Decision Letter · Decision Letter 0]

20 Apr 2022

PONE-D-22-07436Characterization of a mutant lacking a phage-derived gene region from Treponema denticola ATCC 35405PLOS ONE

Dear Dr. Nagano,

Thank you for submitting your manuscript to PLOS ONE. After careful consideration, we feel that it has merit but does not fully meet PLOS ONE’s publication criteria as it currently stands. Therefore, we invite you to submit a revised version of the manuscript that addresses the points raised during the review process. It will be fine to keep your data and discussions of dentilisin and bacterial length. Please ensure that your decision is justified on PLOS ONE’s publication criteria and not, for example, on novelty or perceived impact.

We look forward to receiving your revised manuscript.

Kind regards,

Brian Stevenson, Ph.D.

Academic Editor

PLOS ONE

Journal Requirements:

Reviewers' comments:

Reviewer's Responses to Questions

**Comments to the Author**

1. Is the manuscript technically sound, and do the data support the conclusions?

Reviewer #1: Yes

Reviewer #2: Partly

Reviewer #3: No

2. Has the statistical analysis been performed appropriately and rigorously? 

Reviewer #1: Yes

Reviewer #2: Yes

Reviewer #3: No

3. Have the authors made all data underlying the findings in their manuscript fully available?

Reviewer #1: Yes

Reviewer #2: No

Reviewer #3: Yes

4. Is the manuscript presented in an intelligible fashion and written in standard English?

Reviewer #1: Yes

Reviewer #2: Yes

Reviewer #3: No

5. Review Comments to the Author

Reviewer #1: This manuscript reports on characterization of a spontaneous mutant of Treponema denticola ATCC35405 that appears to have resulted from multiple passages in culture. Genomic sequencing of the mutant strain is showed that it lost a 40-kb region that includes a prophage that was previously described by others, as well as portions of at least two other distally located genes. The mutant 35405 strain differed significantly from a lower-passage 35405 strain in total growth, motility and time-dependent adherence to epithelial cells. This manuscript is of interest specifically because it highlights the potential reliability risks of working with high-passage laboratory strains. However, due to the significant documented genomic differences between the two strains, it will likely be difficult to specifically determine the importance of each missing gene. There are several areas that should be improved, particularly in documentation and description of methodology. The absence of citation and discussion of previous work characterizing the T. denticola prophage is a serious omission that must be corrected.

Specific comments:

1. Abstract, first line: The statement that T. denticola "is responsible for periodontal disease" is much too strong.

2. Line 43: "In contrast, T. denticola..." In contrast to what, specifically?

3. Motility assay, lines 98-100: Please provide an appropriate citation for video determination of rotation rate.

4. Line 147: There appears to be a typo here? "...visual field of 130^2 μm^2"

5. Genome analysis and Table 1: The authors must cite and discuss the report by Mitchell et al. (2010) (DOI 10.1099/mic.0.033654-0774 033654) that characterized the prophage region of the genome of T. denticola ATCC 35405 (TDE1133-TDE1173) and showed that T. denticola produced bacteriophage particles and a circularized bacteriophage genome. This is a major oversight that seriously detracts from the rigor of this manuscript.

6. Line 249: Please define "association number."

7. Fig. 3: To more accurately portray the differences in cell adherence of the two strains, it would be appropriate to include images taken after 3h incubation. The image shown (1h) is not "representative" of the differences discussed.

8. Lines 299-300: Please see comment #5 above regarding the T. denticola phage.

9. Lines 332-333: The two studies cited on the relationship between growth rate and flagellar gene expression in E. coli seem to come to conflicting conclusions.

10. Line 338: "strongly involved" is quite an overstatement here. Differences were observed, but the mechanisms involved were not determined in the cited study.

Reviewer #2: The manuscript by Yokogawa et al. characterized a laboratory-evolved strain of the spirochete Treponema denticola (ATCC . The authors note that compared to the ancestral strain (called OG for original), the evolved strain (called MT for mutant) grew better in vitro (both liquid and solid media), had decreased motility, and displayed reduced adhesion to epithelial cells. The authors provide genomes for both the OG and MT strains, which represent a nice resource for the research community. I have some comments and clarifications for the authors to consider.

1. The authors previous work (Comparative analysis of motility and other properties of Treponema denticola strains, PMID: 27914958) suggests that T. denticola ATCC 35405 has low motility due to culture methods and the authors point this out here in lines 52-53: “In a previous study, we attributed the decrease in motility of ATCC 35405 to our culture method, which we developed using a novel commercially based medium.” In this study, the authors “report that the low motility strain derived from ATCC 35405 is a mutant lacking a possible phage-derived gene region from the original strain.” (lines 56-57). Does this warrant a correction be issued for the authors previous work?

2. How does incubating epithelial cells under anaerobic conditions for 3 hours affect their viability and how might this affect the bacterial adhesion results? Additional experiments are not necessary, but this caveat should be pointed out to readers.

3. Was the RNAseq data uploaded to GenBank? If not, it should be. Also, the authors should include the RNAseq analysis as a supplementary table that shows stats, names, fold change etc. for all genes.

4. The authors suggest that the prophage may have been excised from the chromosome. In the OG strain, are there direct repeats flanking the deleted prophage and is one of these repeats present in the MT strain? If so, this would indicate natural excision of the prophage. If the repeat is missing in the MT strain (i.e., the att site is gone), then the prophage would not be able to re-integrate back into the chromosome and may help explain the loss of this region.

5. The methods indicate electron microscopy was performed. I would love to see some representative images included in a resubmission. Did these images show periplasmic flagella in both strains? The presence/absence of flagella would be useful to know in light of the motility and epithelial cell adhesion data the authors present.

6. Line 147: Please remove the first superscript: 130^2 cm^2.

7. Line 267: “MT showed statistically significantly higher and lower transcriptional activity…(Table 5)” I do not see any statistics in Table 5. Can the authors please add these values? I prefer to visualize such data as a volcano plot, but this is only my personal preference.

8. Line 290: remove the “4” before the word discussion.

9. Line 294: “No other regions where phage-related genes accumulated were found…” I assume the authors used Illumina short-read sequencing (this should be clarified in the Methods). Mapping short-reads back to the ancestral reference genome would likely not pick up additional sequences not present in the reference genome. Thus, the authors cannot make this claim unless they performed long-read sequencing or did some other type of analysis using the reads that did not map back to the reference genome. Can the authors please clarify or remove this statement and other relevant discussion related to it?

10. Line 330: “Unfortunately, we did not clarify the genetic background leading to a decrease in motility of MT.” Did the authors look at SNPs, indels, and other types of mutations? These could potentially explain the loss of motility/other phenotypes.

Reviewer #3: The finding that 35405 has lost segments of its genome is interesting and worth noting. However, the paper is compromised by its delivery and by the inclusion of weak or irrelevant data. The most significant finding appears to be the loss of phage but that is not discussed in the “discussion section”. The authors should talk about what is known regarding phage in T. denticola. The paper should be streamlined considerably.

Did the genome sequencing identify any point mutations etc or were the only differences the deletions.

Regarding the dentilisin assays and cell length analyses, there is no need to list the methods or show the data since there were no significant differences in these properties. In fact, it is not clear why dentilisin activity was looked at in the first place. In addition, there are no controls such as a dentilisin deficient strain (33521). The rationale for why each experiment was done should be discussed. The methods, associated tables and figures for cell length and dentilisin activity should be dropped. The information could be briefly mentioned in the text.

The discussion about motility requires clarification – the authors need to consider both rotational and translational motility. They refer only to rotational motility but I think they mean translational. Also, is appears that motility was assessed with cells that had already reached the stationary phase. Hence some of the cells may be dead in the MT strain. If this is the case, then the assays are not valid.

The term “pathogenic characters” needs to be rephrased

The term “transcriptional activity” is incorrectly used. The authors seem to suggest that they are assessing the overall transcriptional activity of the cell but they only talk about select genes. It seems as though the transcriptional analyses are not be correctly assessed. There is presumably a lot of information in the RNA seq analyses that could be focused on. In addition, the differential expression of individual genes was not validated using an independent approach. It would also have been useful if they mentioned what kind of numbers they saw for the genes that were apparently deleted. Did the RNA seq yield the expected results for these genes.

Are the colonies’ images presented in figure 2 from the sample plate or is that a merged image?

6. PLOS authors have the option to publish the peer review history of their article (what does this mean?). If published, this will include your full peer review and any attached files.

Reviewer #1: No

Reviewer #2: No

Reviewer #3: No

---

## [Author Response · Author response to Decision Letter 0]

20 May 2022

Response to Reviewer #1

Thank you for your insightful comments. We profoundly appreciate your letting us know about the paper from Mitchell et al. We cited the paper and improved the introduction, results and discussion sections in the revised version. 

As you indicated, we did not identify the gene in association with a characteristic difference such as growth and motility. However, since the DNA region identified in this study probably influence the alteration in phenotype, we believe that our study contains useful information for the development of research in this field. 

Below are the answers to the specific comments from you.

1. Abstract, first line: The statement that T. denticola "is responsible for periodontal disease" is much too strong.

Changed it into “associated with”.

2. Line 43: "In contrast, T. denticola..." In contrast to what, specifically?

Removed it. Although we used “In contrast” to contrast motility with biofilm formation, we agree that the phrase makes a confusion to readers.

3. Motility assay, lines 98-100 (should be 89-96): Please provide an appropriate citation for video determination of rotation rate.

Add a reference of our previous paper.

4. Line 147: There appears to be a typo here? "...visual field of 130^2 μm^2"

We modified it to be “130 μm × 130 μm (= 16,900 μm^2). In addition, we corrected it in the Fig. 3 legend.

5. Genome analysis and Table 1: The authors must cite and discuss the report by Mitchell et al. (2010) (DOI 10.1099/mic.0.033654-0774 033654) that characterized the prophage region of the genome of T. denticola ATCC 35405 (TDE1133-TDE1173) and showed that T. denticola produced bacteriophage particles and a circularized bacteriophage genome. This is a major oversight that seriously detracts from the rigor of this manuscript.

We corrected and improved the introduction, results and discussion sections with citing the paper from Mitchell et al. In addition, we moved the table 1 of the phage-related genes in the first version into the supporting information because it has been described by Mitchel et al.

6. Line 249: Please define "association number."

Added the description of "association number" in the materials and methods section. We could not distinguish the bacterial cells between adherence on the cell surface and invasion into the cell, and we expressed the bacterial counts as the association number including both adherence and invasion.

7. Fig. 3: To more accurately portray the differences in cell adherence of the two strains, it would be appropriate to include images taken after 3h incubation. The image shown (1h) is not "representative" of the differences discussed.

The images were of 3 h. We apologize for our mistake.

8. Lines 299-300: Please see comment #5 above regarding the T. denticola phage.

Corrected with citing the paper as described above.

9. Lines 332-333: The two studies cited on the relationship between growth rate and flagellar gene expression in E. coli seem to come to conflicting conclusions.

These papers were cited as examples of effect of bacterial conditions during the growth on motility. We do not discuss the conclusions of these two papers.

10. Line 338: "strongly involved" is quite an overstatement here. Differences were observed, but the mechanisms involved were not determined in the cited study.

Changed “strongly” to “possibly”.

 

Response to Reviewer #2

Thank you for your positive and constructive comments. According to your comment, we examined a point mutation such as SNPs, and found that more than 200 mutations were accumulated in the MT strain. Therefore, we have thoroughly revised our manuscript including the title. The answers to each point from you are given below. 

1. The authors previous work (Comparative analysis of motility and other properties of Treponema denticola strains, PMID: 27914958) suggests that T. denticola ATCC 35405 has low motility due to culture methods and the authors point this out here in lines 52-53: “In a previous study, we attributed the decrease in motility of ATCC 35405 to our culture method, which we developed using a novel commercially based medium.” In this study, the authors “report that the low motility strain derived from ATCC 35405 is a mutant lacking a possible phage-derived gene region from the original strain.” (lines 56-57). Does this warrant a correction be issued for the authors previous work?

We will ask the journal (Microbial Pathogenesis) to announce a correction comment, when this manuscript is published.

2. How does incubating epithelial cells under anaerobic conditions for 3 hours affect their viability and how might this affect the bacterial adhesion results? Additional experiments are not necessary, but this caveat should be pointed out to readers.

Our previous study (reference # 5) showed that the cell viability reagent (alamarBlue) did not detect a decrease in cell viability over 24 hours.

3. Was the RNAseq data uploaded to GenBank? If not, it should be. Also, the authors should include the RNAseq analysis as a supplementary table that shows stats, names, fold change etc. for all genes.

Thank you for your valuable suggestions. We deposited the RNAseq data with GenBank, and described it in the materials and methods section. In addition, all data of RNAseq analysis are added to Supporting Information.

4. The authors suggest that the prophage may have been excised from the chromosome. In the OG strain, are there direct repeats flanking the deleted prophage and is one of these repeats present in the MT strain? If so, this would indicate natural excision of the prophage. If the repeat is missing in the MT strain (i.e., the att site is gone), then the prophage would not be able to re-integrate back into the chromosome and may help explain the loss of this region.

Thank you for your important suggestions. The reviewer #1 let us know about the paper on the phage induction in this strain (ATCC 35405). Then we found the predicted att site, and described it in the results and discussion sections.

5. The methods indicate electron microscopy was performed. I would love to see some representative images included in a resubmission. Did these images show periplasmic flagella in both strains? The presence/absence of flagella would be useful to know in light of the motility and epithelial cell adhesion data the authors present.

Thank you for your constructive suggestions. Flagella are also observed in the mutant. We added TEM images to Supporting Information.

6. Line 147: Please remove the first superscript: 130^2 cm^2.

We improved it to be “130 μm × 130 μm (= 16,900 μm^2).

7. Line 267: “MT showed statistically significantly higher and lower transcriptional activity…(Table 5)” I do not see any statistics in Table 5. Can the authors please add these values? I prefer to visualize such data as a volcano plot, but this is only my personal preference.

All RNAseq data and its statistical analysis were added to Supporting Information. We also published the raw data of the RNAseq in Genbank, and added the description of it in the revised version.

8. Line 290: remove the “4” before the word discussion.

Removed it. Thank you.

9. Line 294: “No other regions where phage-related genes accumulated were found…” I assume the authors used Illumina short-read sequencing (this should be clarified in the Methods). Mapping short-reads back to the ancestral reference genome would likely not pick up additional sequences not present in the reference genome. Thus, the authors cannot make this claim unless they performed long-read sequencing or did some other type of analysis using the reads that did not map back to the reference genome. Can the authors please clarify or remove this statement and other relevant discussion related to it?

Improved the sentence. We would like to describe that accumulation of phage-related genes were not seen in the genome of the ATCC 35405 (original strain) published in GenBank.

10. Line 330: “Unfortunately, we did not clarify the genetic background leading to a decrease in motility of MT.” Did the authors look at SNPs, indels, and other types of mutations? These could potentially explain the loss of motility/other phenotypes.

Thank you for your valuable comment. We listed them in the supporting information file, and improved the text.

 

Response to Reviewer #3

We appreciate your crucial comments. According to your comment, we investigated mutations such as SNPs, and found that many mutations were accumulated in the MT strain. Therefore, we have thoroughly revised our manuscript including the title. The answers to each point from you are given below.

1. The most significant finding appears to be the loss of phage but that is not discussed in the “discussion section”. The authors should talk about what is known regarding phage in T. denticola. The paper should be streamlined considerably.

Thank you very much for your crucial indication. Reviewer #1 also requires to cite the paper about a phage induction in T. denticola ATCC 35405 from Mitchel et al. We cited the paper, and improved the introduction, results and discussion sections.

2. Did the genome sequencing identify any point mutations etc or were the only differences the deletions.

Thank you for your valuable comment. We listed them in the supporting information file, and improved the text.

3. Regarding the dentilisin assays and cell length analyses, there is no need to list the methods or show the data since there were no significant differences in these properties. In fact, it is not clear why dentilisin activity was looked at in the first place. In addition, there are no controls such as a dentilisin deficient strain (33521). The rationale for why each experiment was done should be discussed. The methods, associated tables and figures for cell length and dentilisin activity should be dropped. The information could be briefly mentioned in the text.

We would like to leave the descriptions and tables regarding the dentilisin and cell length analyses. We think that the information of cell length and dentilisin is important in T. denticola study even if they were not significantly different between the strains.

4. The discussion about motility requires clarification – the authors need to consider both rotational and translational motility. They refer only to rotational motility but I think they mean translational. Also, is appears that motility was assessed with cells that had already reached the stationary phase. Hence some of the cells may be dead in the MT strain. If this is the case, then the assays are not valid.

We used bacteria in the logarithmic growth phase as described in the bacterial strain and culture part of the materials and methods section.

Spirochete does not move forward in a liquid medium with low viscosity. Therefore, we examined rotational rate in this study. Some papers have examined the bacterial speed using a viscous medium, but we have not yet been able to perform the experiment with reproducibility.

5. The term “pathogenic characters” needs to be rephrased

We are sorry not to understand the point in your comment. However, we unified the sections of physiologic and pathogenic characters because we reconsidered that they were not clearly distinguished. In addition, we deleted unnecessary sentences at the beginning of each chapter.

6. The term “transcriptional activity” is incorrectly used. The authors seem to suggest that they are assessing the overall transcriptional activity of the cell but they only talk about select genes. It seems as though the transcriptional analyses are not be correctly assessed. There is presumably a lot of information in the RNA seq analyses that could be focused on. In addition, the differential expression of individual genes was not validated using an independent approach. It would also have been useful if they mentioned what kind of numbers they saw for the genes that were apparently deleted. Did the RNA seq yield the expected results for these genes.

All data of RNAseq and statistical analysis were posted in the supporting information, and the raw sequence data were also published in Genbank.

We added a description that RNAseq analysis was performed for all genes, and we showed only the genes that the expression ratio was more than twice with statistically significant difference between the strains, although we have mentioned it in the margin of Table 5.

The expression ratios of the lacking genes were 18 to 10,000 times or more by statistical calculation, consisting with the deletion of the DNA region.

7. Are the colonies’ images presented in figure 2 from the sample plate or is that a merged image?

We think that you would like to describe “same plate”. If so, it was the same plate.

---

## [Editor Report · Decision Letter 1]

7 Jun 2022

Characterization of a Treponema denticola ATCC 35405 mutant strain with mutation accumulation, including a lack of phage-derived genes

PONE-D-22-07436R1

Dear Dr. Nagano,

We’re pleased to inform you that your manuscript has been judged scientifically suitable for publication and will be formally accepted for publication once it meets all outstanding technical requirements.

Kind regards,

Brian Stevenson, Ph.D.

Academic Editor

PLOS ONE
---

## [Editor Report · Acceptance letter]

16 Jun 2022

PONE-D-22-07436R1 

Characterization of a *Treponema denticola* ATCC 35405 mutant strain with mutation accumulation, including a lack of phage-derived genes 

Dear Dr. Nagano:

I'm pleased to inform you that your manuscript has been deemed suitable for publication in PLOS ONE. Congratulations! Your manuscript is now with our production department. 

Kind regards, 

on behalf of

Prof. Brian Stevenson 

Academic Editor

PLOS ONE